# Variational Mode Decomposition Analysis of Electroencephalograms during General Anesthesia: Using the Grey Wolf Optimizer to Determine Hyperparameters

**DOI:** 10.3390/s24175749

**Published:** 2024-09-04

**Authors:** Kosuke Kushimoto, Yurie Obata, Tomomi Yamada, Mao Kinoshita, Koichi Akiyama, Teiji Sawa

**Affiliations:** 1Department of Anesthesiology, Graduate School of Medical Science, Kyoto Prefectural University of Medicine, Kyoto 602-8566, Japan; kushihai@koto.kpu-m.ac.jp (K.K.); t5omomi@koto.kpu-m.ac.jp (T.Y.); mao6515@koto.kpu-m.ac.jp (M.K.); 2Department of Anesthesiology, Yodogawa Christian Hospital, Osaka 533-0024, Japan; 3120033@ych.or.jp; 3Department of Anesthesiology, Kindai University, Higashiosaka 577-8502, Japan; kanaslik@koto.kpu-m.ac.jp; 4Hospital of Kyoto Prefectural University of Medicine, Kyoto 602-8566, Japan

**Keywords:** electroencephalogram, general anesthesia, grey wolf optimizer, Hilbert–Huang transform, variational mode decomposition

## Abstract

Frequency analysis via electroencephalography (EEG) during general anesthesia is used to develop techniques for measuring anesthesia depth. Variational mode decomposition (VMD) enables mathematical optimization methods to decompose EEG signals into natural number intrinsic mode functions with distinct narrow bands. However, the analysis requires the a priori determination of hyperparameters, including the decomposition number (K) and the penalty factor (PF). In the VMD analysis of EEGs derived from a noninterventional and noninvasive retrospective observational study, we adapted the grey wolf optimizer (GWO) to determine the K and PF hyperparameters of the VMD. As a metric for optimization, we calculated the envelope function of the IMF decomposed via the VMD method and used its envelope entropy as the fitness function. The K and PF values varied in each epoch, with one epoch being the analytical unit of EEG; however, the fitness values showed convergence at an early stage in the GWO algorithm. The K value was set to 2 to capture the α wave enhancement observed during the maintenance phase of general anesthesia in intrinsic mode function 2 (IMF-2). This study suggests that using the GWO to optimize VMD hyperparameters enables the construction of a robust analytical model for examining the EEG frequency characteristics involved in the effects of general anesthesia.

## 1. Introduction

The decoding technology used to monitor electroencephalograms (EEGs) during general anesthesia (GA) and extract the information required to understand the depth of anesthesia or the state of consciousness is crucial not only for comprehending brain function during GA but also for assessing the appropriateness of the anesthesia level [1,2,3]. There have been numerous reports on new technologies for EEG signal processing that can significantly broaden the scope of EEG monitoring for assessing patients’ anesthetic sedation states during GA [4,5]. In addition, there are growing expectations for advancements in the intelligent sensors that are involved in detecting biological signals, spurred by the rapid growth of flexible electronics [6,7]. The potential benefits of these technologies, when combined with fast AI diagnostic algorithms, are significant. This combination holds promise for more reliable clinical diagnostics, especially in the analysis of EEGs. However, the analysis parameters of commercially available EEG monitors have only been optimized for certain inhaled anesthetics and propofol, and other anesthetics or the effects of age have not been accounted for [8,9,10]. Additionally, each algorithm is proprietary and not disclosed, which limits comparative studies and further developments. Therefore, establishing new technologies for effective feature extraction of EEGs during GA using open algorithms is crucial for anesthesiologists who aim to provide safe and effective anesthesia.

Mode decomposition methods have emerged as a new approach to feature extraction, with research on these methods beginning at the end of the 20th century [11]. We have examined and assessed the usefulness of various mode decomposition methods for extracting features from EEG signals under GA [12,13]. The origin of mode decomposition is the empirical mode decomposition (EMD) method of Huang et al. [11], which analyzes nonstationary signals, and the use of EMD to analyze EEG waves during GA has been reported. However, EMD has the drawback that it decomposes data into intrinsic mode functions (IMFs) without band limitations, leading to frequency mixing issues and difficulties in result interpretation. In 2013, Dragomiretsky and Zosso proposed variational mode decomposition (VMD) [14], a method for overcoming the shortcomings of EMD. VMD views the decomposition of signals as a variational problem, solving a functional equation by applying mathematical optimization algorithms. This procedure continuously updates the central frequency of each mode online in the frequency domain, decomposing the signals into multiple narrowband IMFs. However, the VMD method heavily depends on its hyperparameters, namely, the mode decomposition number (K) and the penalty factor (PF), which affect the accuracy of capturing useful modality information features. Determination of the K value requires some prior knowledge of the original signal, and the bandwidth of each modality obtained through VMD is determined by the PF, with a larger PF narrowing the bandwidth. Implementing the VMD method for EEG feature extraction necessitates the careful selection of the acceptable values for these hyperparameters.

Recently, the grey wolf optimizer (GWO) algorithm, a new heuristic algorithm for optimization problems, was proposed to solve this issue [15,16,17]. The GWO algorithm, developed by Mirjalili [18], is a swarm intelligence optimization method that mimics the hunting behavior and leadership hierarchy of grey wolves, and it has been successful in many real-world applications [19]. In this study, we used envelope entropy as the fitness function for the GWO algorithm, which we used to optimize the VMD method by determining the number of K modes and the PF in EEG signals during GA. 

The signals analyzed are complex and volatile, making it challenging to determine the two parameters influencing them. Therefore, selecting appropriate parameters is key to analyzing signal data through VMD. Attempts to optimize VMD using intelligent algorithms have been reported in various fields. It is essential to use an algorithm that is easy to implement, has an adaptive convergence coefficient and information feedback mechanisms, and balances local optimization with global search. The GWO algorithm, which mimics the hunting behavior and leadership hierarchy of grey wolves, has been successful in many real-world applications. In the same number of iterations, the GWO algorithm has been reported to converge significantly better than other algorithms. Therefore, in this study, we implemented the GWO algorithm in the EEG analysis process with the aim of optimizing VMD. To do this, we used envelope entropy as the fitness function of the GWO algorithm to optimize the number of K modes and the PF in EEG signals during GA. As the effects of GA drugs diminish, there are changes in the frequency components of EEG signals. In such dynamically changing EEG signals, it is assumed that the hyperparameters of VMD also need to change dynamically. Based on this hypothesis, we conducted an experiment where the hyperparameters of VMD were calculated in real time for each EEG analysis epoch to optimize them. Specifically, during the approximately 10 min from the maintenance of GA to awakening, we observed how VMD’s hyperparameters were determined via the GWO algorithm and how the associated VMD process was optimized. We also examined whether the extraction of EEG features using VMD could lead to the estimation of anesthesia depth through the acquisition of IMF components and the analysis of their characteristic frequency components. In this study, we report on the use of an optimized VMD method for decomposing EEG data recorded under GA, demonstrating how it can be used to acquire a series of IMF components and examining its utility.

## 2. Materials and Methods

### 2.1. Anesthesia Management and Data Acquisition

All experimental protocols involving human participants adhered to the principles outlined in the Declaration of Helsinki. The current study (No. ERB-C-1074) received approval from the Institutional Review Board (IRB) for human experiments at the Kyoto Prefectural University of Medicine (KPUM). For this noninterventional and noninvasive retrospective observational study, the need for informed patient consent was waived by the KPUM IRB. Nevertheless, patients were informed of their opportunity to opt out and were provided this opportunity during their preoperative anesthesia clinic visit. The EEG data recorded from three patients who were treated with total anesthesia using propofol (for 10 min before waking from GA) were used for analysis (Appendix A). The EEG dataset used is available on the author’s GitHub site (https://github.com/teijisw/EEG_DataSet, accessed on 4 July 2024) [20].

Anesthesia was induced with small doses of fentanyl (1 µg kg^−1^ per dose) and a continuous intravenous infusion of remifentanil (0.125–0.25 µg kg^−1^ minute^−1^), followed by a single intravenous injection of propofol (2 mg kg^−1^) and rocuronium (0.8–1.0 mg kg^−1^). Tracheal intubation was performed, and anesthesia was maintained with propofol using a target-controlled infusion pump (Terufusion™ TIC pump TE-371, Terumo, Tokyo), aiming for a blood concentration of 3 µg/mL. Additional maintenance doses of fentanyl (1 µg/kg per dose) and rocuronium (0.2 mg/kg at 20–30-minute intervals), along with a continuous intravenous infusion of remifentanil (0.125–0.25 µg/kg per minute), were administered. To acquire EEG data, we utilized the EEG Analyzer software (ver. 54_GP; available at http://anesth-kpum.org/blog_ts/?p=3169 4 July 2024) [13,21]. Raw EEG signals were recorded as text files on a personal computer via the RS-232 interface of a VISTA A-3000 BIS monitor (32-bit raw EEG signals at a sampling frequency of 128 Hz, eight packets/s, VISTA Application revision 3.22, Platform revision 2.03, Medtronic, Minneapolis, MN, USA) with a BIS Quatro sensor mounted on the frontal regions, as previously reported [13].

VMD and GWO programming codes were constructed in the computer programming languages Python (ver.3.8) and Processing (ver.4.3), with the Apache Common Mathematics Library (https://commons.apache.org/proper/commons-math/, 4 July 2024, ver.3.6.1), Digital Signal Processing in Java (JDSP ver.0.5.0, https://jdsp.dev 4 July 2024), and FloatTable.pde from Example for Visualizing Data by Ben Fry (https://benfry.com/writing/archives/3/ 4 July 2024), as shown in Appendix A.

### 2.2. Algorithm for VMD and the Hilbert Transform

VMD divides an input signal *f* into k narrowband IMFs *u_k_* characterized by a discrete central frequency *ω_k_* with a particular sparsity property [14]. The following three mathematical steps are involved in the VMD algorithm [14]:1.The first step computes the analytic signal associated with each IMF according to the Hilbert transform (Formula (1)).
(1)$$\;\;\left(\delta \left(t\right)+\frac{j}{\pi t}\right)*u_{k}\left(t\right)$$
where * is the convolution, *u*_k_(*t*) is the k^th^ IMF, and δ(*t*) is the pulse signal.

2.The second step involves demodulating the analytic signal to the baseband by multiplying it with an exponential function, which is adjusted to the respective estimated central frequency $\omega_{k}$.
(2)$$\partial_{t}\left[\left(\delta \left(t\right)+\frac{j}{\pi t}\right)\times u_{k}\left(t\right)\right]e^{-j\omega_{k}t}$$

3.The third step estimates the bandwidth by using the $H^{1}$ Gaussian smoothness of the demodulated signal, which involves calculating the squared *L*^2^ norm of the slope of the demodulated analytic signal. It then computes the associated analytic signal using the Hilbert transform to obtain the one-sided frequency spectrum of each intrinsic mode *u*_k_.

The constrained variational evaluation of a given signal *x*(*t*) is described as
(3)minuk,ωk⁡∑k∂tδt+jπt×ukte−jωkt22, s.t.∑kuk=f(t),
where *u_k_* and *ω_k_* are shorthand notations for the sets {*u_1_*, ……, *u_k_*} and {*ω_1_*, ……, *ω_k_*}, representing all modes and their central frequencies, respectively. The sum $\sum_{k}:=\sum_{k=1}^{K}$ of all mode functions *u_k_* is equal to the specified time series signal *f*(*t*).

This constrained variational problem is addressed by incorporating a quadratic penalty term and introducing the Lagrange multiplier, λ, into the mathematical optimization method. This methodology effectively transforms the problem into one that involves an unconstrained extended Lagrangian function, denoted as L in Formula (4).
(4)Luk, ωk, λ≔ α ∑k∂tδt+jπt×ukte−jωkt22     +ft−∑kukt22+λt, ft−∑kukt

The solution to this minimization problem is found as a saddle point of the augmented Lagrangian L in a series of iterative partial optimizations called the alternating direction method of multipliers (ADMM, Algorithm 1).
**Algorithm 1. Optimization Concept for VMD [14].**
initialization $\left\{{\hat{u}}_{k}^{1}\right\},\;\left\{{\hat{\omega }}_{k}^{1}\right\},\;{\hat{\lambda }}^{1},\;n\;\leftarrow 0$  repeat    n←n+1   for k=1:K do    Update ${\hat{u}}_{k}$ for all ω ≥0: 
${\hat{u}}_{k}^{n+1}(\omega )\leftarrow \;\;\frac{\hat{f}\left(\omega \right)-\sum_{i<k}{\hat{u}}_{i}^{n+1}\left(\omega \right)-\sum_{i>k}{\hat{u}}_{i}^{n}\left(\omega \right)+\frac{\hat{\lambda }(\omega )}{2}}{1+2\alpha {\left(\omega -\omega_{k}^{n}\right)}^{2}}$(5)    Update ${\hat{\omega }}_{k}:$$\omega_{k}^{n+1}=\frac{\int_{0}^{\infty }\omega {\left|{{\hat{u}}_{k}}_{k}^{n+1}\left(\omega \right)\right|}^{2}d\omega }{\int_{0}^{\infty }{\left|{{\hat{u}}_{k}}_{k}^{n+1}\left(\omega \right)\right|}^{2}d\omega }$(6)  end for   Dual ascent for all ω ≥0:${\hat{\lambda }}^{n+1}\;\leftarrow \;{\hat{\lambda }}^{n}+\tau \left(\hat{f}(\omega )-\sum_{k}{\hat{u}}_{k}^{n+1}\;(\omega )\right)\;\;\;\;$(7)until convergence:∑ku^kn+1−u^kn 22u^kn 22<ϵ(8)

For an EEG signal *x*(*t*), VMD decomposes it into a series of IMFs, C*n* (where *n* = 1, 2, …, *N*), with *N* being the total number of IMFs. After VMD, the signal *x*(*t*) can be expressed as
(9)$$x\left(t\right)=\sum_{i=1}^{N}{imf(t)}_{u}$$

Applying the Hilbert transform to the IMF components, it follows that
(10)$$Z\left(t\right)=imf\left(t\right)+iH\left[imf\left(t\right)\right]=a\left(t\right)e^{i\int \omega \left(t\right)dt}$$
in which
(11)$$a\left(t\right)=\sqrt{{imf}^{2}\left(t\right)+H^{2}[\mathrm{imf}\left(t\right)]}$$
(12)$$\omega \left(t\right)=\frac{d}{dt}\left[artan\left\{\frac{H\left[imf\left(t\right)\right]}{imf\left(t\right)}\right)\right]$$
(13)$$h\left(\omega \right)=\int H\left(\omega ,\;t\right)dt$$
where *a*(*t*) is the amplitude of the IMF and ω(*t*) is the instantaneous frequency for obtaining a time–frequency distribution for signal *x*(*t*) and the Hilbert amplitude spectrum *H*(*x*, *t*).

### 2.3. Grey Wolf Optimizer

In the original VMD method, users need to provide the number of K modes and the PF in advance, although there is no guarantee whether these parameters are optimal or whether they will achieve satisfactory results [21,22]. The smaller the PF, the larger the bandwidth of each IMF component. Furthermore, improper selection of the number of K modes can directly lead to signaling component distortion and unrealistic denoising effects. In practice, the two influencing parameters are usually difficult to determine because the actual signal to be analyzed is complex and changeable. Thus, selecting suitable parameters is key to the VMD method. Next, we introduce the principle of the GWO algorithm (Figure 1) and its convergence and the steps of the parameter-optimized VMD method.

The GWO is an optimization algorithm simulating the social hierarchical relationships and hunting behaviors of grey wolves in nature [18]. To mathematically model the social hierarchy of wolves, the population (a candidate group of hyperparameter values to be optimized) corresponding to a pack of wolves is divided into four social classes, and the optimal solution is considered to be an alpha (α) wolf. The second and third best solutions are treated as the beta (β) and delta (δ) wolves, respectively, and the remaining candidate solutions in the swarm are allocated to the omega (ω) wolves. In the GWO algorithm, hunting (optimization) is guided by the top three solutions—α, β, and δ—with the ω wolf following these three wolves. Although it is unclear during the analysis where the optimal solution will be, it is assumed that a possible optimal solution is surrounded by three optimal solutions and that the movement of the ω wolf population approaching the optimal solutions (α, β, and δ) is randomly controlled. We adopted a method that approaches the optimal solution by incorporating and repeating variables, introducing a fitness function, and sequentially replacing the α, β, and δ wolves in the population.

Convergence of the GWO algorithm: Grey wolves surround their prey when hunting. To mathematically model this enveloping behavior, the following equation is proposed (Figure 2).
(14)$$\vec{D}=\left\lceil \vec{C}\cdot \vec{X_{p}}(i)-\vec{X}(i)\right\rceil$$
(15)$$\;\vec{X}(i+1)=\left\lceil \vec{X_{p}}(i)-\vec{A}\cdot \vec{D}\right\rceil$$
where *i* indicates the current iteration, $\vec{A}$ and $\vec{C}$ are the coefficient vectors, $\vec{X_{p}}$ is the position vector of the prey, and $\vec{X}$ indicates the position vector of a grey wolf.

The vectors $\vec{A}$ and $\vec{C}$ are calculated as follows:(16)$$\vec{A}=\vec{a}\cdot \left\lceil 2\cdot \vec{r_{1}}-1\right\rceil$$
(17)$$\vec{C}=2\cdot \vec{r_{2}}$$

Here, the components of $\vec{a}$ in Formula (16) decrease linearly from 2 to 0 throughout the iterations, and $\vec{r_{1}}$ and $\vec{r_{2}}$ in Formulas (16) and (17) are random vectors in [0, 1]. The coefficient vector $\vec{A}$ of Formula (15) is related to the distance $\vec{D}$ between each wolf $\vec{X}(i)$ and the prey $\vec{X_{p}}$. However, the position of the prey $\vec{X_{p}}$ itself is also multiplied by a coefficient vector $\vec{C}$ within a random range of [0, 2] in Formula (14), providing random weight to the prey by either emphasizing ($\vec{C}$ > 1) or de-emphasizing ($\vec{C}$ < 1) it. This allows GWO to exhibit more random behavior throughout the optimization, facilitating exploration and avoidance of local optima. The random variation in $\vec{A}$, which initially ranges from −2 to +2, decreases gradually with each loop, forcing the search agents also to move away from the prey. This emphasis on exploration enables the GWO algorithm to search globally. To mathematically simulate the hunting behavior of grey wolves, we need to know the location of the prey. However, we cannot actually know it directly. Therefore, GWO assumes that α, β, and δ are related to the potential location of the prey. In other words, it is assumed that the prey position (optimal solution) is surrounded by three wolves, α, β, and δ. The first three best solutions obtained thus far are preserved, and it is mandated that the other search agents (including ω) update their positions according to the positions of the best search agents. In this regard, the following formula is proposed. In this case, the positions of wolves α, β, and δ are used instead of the prey position ($\vec{X_{p}}$) in Formulas (14) and (15), and the ω wolf position is updated in three ways using the following formula.
(18)$$\vec{D_{\alpha }}=\left\lceil \vec{C_{1}}\cdot \vec{X_{\alpha }}-\vec{X}\right\rceil ,\;\vec{D_{\beta }}=\left\lceil \vec{C_{2}}\cdot \vec{X_{\beta }}-\vec{X}\right\rceil ,\;\vec{D_{\delta }}=\left\lceil \vec{C_{3}}\cdot \vec{X_{\delta }}-\vec{X}\right\rceil$$
(19)$$\vec{X_{1}}=\left\lceil \vec{X_{\alpha }}-\vec{A_{1}}\cdot \vec{D_{\alpha }}\right\rceil ,\;\vec{X_{2}}=\left\lceil \vec{X_{\beta }}-\vec{A_{2}}\cdot \vec{D_{\beta }}\right\rceil ,\;\vec{X_{1}}=\left\lceil \vec{X_{\alpha }}-\vec{A_{1}}\cdot \vec{D_{\alpha }}\right\rceil$$

The center of the three possible updated positions is selected as a new ω wolf update position candidate, as follows:(20)$$\vec{X}(i+1)=\frac{\vec{X_{1}}+\vec{X_{2}}+\vec{X_{3}}}{3}$$

However, the position will be updated only when the fitness function shows better fitting, and after updating the positions of the wolves in the entire pack using this method, the top three wolves showing the best values will be updated using the fitness function. By repeating the task of specifying the α, β, and δ wolves a specified number of times, we hope that most of the wolves will converge to the optimal solution.

When using these equations for VMD, the search agent updates its position in the two-dimensional search space for K and PF according to α, β, and δ (Figure 3). Furthermore, the final position is a random location within a circle defined by the positions of α, β, and δ in the search space. In other words, α, β, and δ estimate the position of the prey, and other wolves randomly update their positions around the prey. Through the fitness function, the positions of the wolves ranked again are used to update α, β, and δ, and by repeating this loop, the solution offering an optimized fitness function (a combination of optimal solutions) is progressively refined.

### 2.4. The Fitness Function: Envelope Entropy

In a situation where only one of several events occurs, and each can occur with equal probability *P*, the information *I* gained upon learning which event actually occurred can be expressed as
(21)$$I={log}_{2}\frac{1}{P}=-{log}_{2}P$$

Therefore, when this process is repeated *i* times and the probability of an event occurring is $P_{i}$, the information obtained when it occurs is ${-log}_{2}P_{i}$. From this, the average amount of information obtained is
(22)$$I=-\sum_{i}P_{i}\times {log}_{2}P_{i}\;$$

The information entropy *S* before the outcome is known can also be determined using the same formula.
(23)$$S=-\sum_{i}P_{i}\times {log}_{2}P_{i}\;$$

Here, if $B\left(i\right)\;$ is the envelope signal of *x*(*i*), then $B\left(i\right)$ can be normalized to obtain the probability $P_{i}$ as follows:(24)$$P_{i}=\frac{B\left(i\right)}{\sum_{i=1}^{N}B(i)}$$

Then, by applying the formula for information entropy to the envelope, we obtain the envelope entropy as follows:(25)$$E_{p}=-\sum_{i=1}^{N}P_{i}\times {log}_{2}P_{i}\;$$

In the GWO algorithm, the envelope entropy of the IMF decomposed via VMD is determined, and K and PF are optimized such that the envelope entropy values of the separated IMFs are reduced (indicating less information). Here, by calculating the envelope entropy of each of the multiple IMFs obtained, the IMF with the highest envelope entropy (the most information, using the maximum value method) is identified. Then, by ensuring that these values become lower (more realistic information) through new combinations of K and PF, it is possible to optimize the separated IMFs overall.

An example using synthetic cosine waves: The following is a synthetic function represented by a combination of three cosine waves with different known frequencies (2 Hz, 12 Hz, and 36 Hz).
(26)$$y=cos(2\times 2\pi t)+\frac{1}{2}cos(12\times 2\pi t)+\frac{1}{4}cos(36\times 2\pi t)$$
where t = [1/1024, 2/1024, …, 1024/1024] (Figure 3A, left). Here, we used this synthetic function to demonstrate an example of the GWO algorithm being used to optimize K and PF. The envelope entropy of this synthetic function was −9.9767506 (Figure 3A, right). If this synthetic function is decomposed with maximum efficiency, it should result in decomposition into cosine waves y=cos(2×2πt), $y=\frac{1}{2}cos(12\times 2\pi t)$, and $y=\frac{1}{4}cos(36\times 2\pi t)$ (Figure 3B, left). Each envelope of the efficiently decomposed components will become a flat line, and the envelope entropy will be −10.0 (Figure 3B, right). This trigonometric function’s composite waveform was analyzed using the GWO algorithm with 20 wolves and 20 iterations, setting K to range from 2 to 6 and PF from 10 to 5000 (randomization in increments of 1 for K and 1 for PF). The envelope entropy of each IMF was calculated, and K and PF were optimized using both the mean value method and the maximum value method. As a result, the GWO algorithm determined the optimal values of K = 3 and PF = 4984. The envelope entropies of the three IMFs were −9.9999944, −9.9999929 (max), and −9.9999958 (Figure 3C). With settings of K = 3 and PF = 10 (Figure 3D) or K = 3 and PF = 70 (Figure 3E), the IMFs were superimposed on the other cosine waves, indicating that complete frequency separation was not achieved.

Then, with PF fixed at the optimized value of 4984 and K set to 2, the application resulted in two frequency components being present in IMF-1, and the envelopes of the IMFs were not flat (Figure 3F). Finally, with PF fixed at the optimized value of 4984 and K set to 4, when VMD was applied, the envelopes of all the IMFs were flat (Figure 3G). However, although the highest envelope entropy was obtained with IMF-4 (−9.9997846), this value was larger than that of IMF-1 (−9.9999944) obtained with GWO, suggesting that GWO with K = 3 produced a better model than VMD with K = 4.

## 3. Results

### 3.1. Convergence Status of K, PF, and Fitness Values in GWO

First, the GWO algorithm optimization process (20 wolves, 20 optimization loops) for the VMD hyperparameters K and PF was performed using one epoch (8 s, 1024 data points) during the maintenance phase of the GA (induced by continuous intravenous propofol administration) in three patients (Appendix A). This involved monitoring the position of each wolf during two-dimensional hunting.

In Patient #1, with 20 optimization loops, convergence to K = 2 occurred at the ninth loop, except for three ω wolves (Figure 4(1)). However, the PF values of the leader wolves, α, β, and δ, converged to 3640, 3620, and 3470, respectively, while the convergence for the ω wolves was unstable. Nonetheless, the fitness function converged to approximately −9.87 to −9.88, indicating that the PF values did not significantly impact the fitness values. In the case of Patient #2 (Figure 4(2)), except for the six ω wolves, convergence to K = 2 occurred in the sixth loop. Similar to Patient #1, the PF values for the α, β, and δ leader wolves were 4900, 4800, and 4680, respectively. The fitness values for these leader wolves had already converged to approximately −9.85 by the second loop, but the fitness function for the ω wolves had not fully converged. In Patient #3 (Figure 4(3)), except for the five ω wolves, there was convergence with K = 3. The PF values of the α, β, and δ leader wolves were 1580, 1830, and 1850, respectively, whereas those of the ω wolves tended to converge to values below 2500. By approximately the fourth loop, the fitness values converged to approximately −9.72 to −9.73. Table 1 shows the convergence status for K, PF, and fitness, with the final convergence value set as 100%. For K, convergence occurred between 6 and 13 loops; for PF and fitness, it was between 9 and 18 loops. Although 20 loops were conducted in this case, sufficient convergence had been achieved by then.

The results show that convergence of the K values was achieved by the α, β, and δ leader wolves at an early stage. However, since the impact on the fitness function was much greater for the K values than for the PF values, it was observed that the convergence of the PF values tended to be delayed. Additionally, when the fitness function converged, the impact of the PF values was considered low.

### 3.2. Temporal Changes in the K, PF, and Fitness Values According to the GWO

Next, for the three cases of GA induced by continuous intravenous propofol administration, the optimal solutions for K and PF were determined through fitness function optimization using the GWO algorithm (20 wolves, 20 optimization loops) for all 73 epochs during the 10 min before and after awakening from GA. Each epoch of EEG data lasted 8 s (128 Hz, 1024 data points).

K was set in the range of 2 to 6 (in increments of 1), and the PF values ranged from 10 to 5000 (in increments of 10). K and PF fluctuated significantly, with K oscillating from a base of 2 up to 3, 4, 5, and 6, while the PF values varied greatly between 10 and 5000 for each epoch (Figure 5A). The Fourier spectrogram for the 10-minute interval was obtained using the multitaper method and is displayed in Figure 5B. Figure 5C shows a Hilbert spectrogram of the IMF of the EEG from approximately the last 10 min before awakening from GA, for K = 2, PF = 2000, and K = 3, PF = 2000. The envelope entropy, excluding the variations after awakening, ranged from approximately −9.8 (Figure 5A). The fitness values derived from the envelope entropy were relatively stable and consistent within a single epoch before awakening. When the fitness values showed a convergence trend, the impact of the PF values on the fitness function appeared low, suggesting that prioritizing the convergence of K would be advisable. After awakening, there were significant increases in fitness, likely influenced by the incorporation of electromyography during the postemergence period, such as orbicularis oculi muscle activity.

### 3.3. VMD Analysis under GWO Support during the Three Phases of General Anesthesia

Finally, we applied the analysis to 8 s EEG segments during the (1) maintenance, (2) transition, and (3) emergence phases of GA induced by propofol (Figure 6A). The power spectrum obtained through Fourier analysis is shown in Figure 6B. The analysis was conducted with 20 wolves, 20 repetitions, and settings of K = 2 to 6 (in increments of 1) and PF = 10 to 5000 (in increments of 10). The results show a convergence to K = 2 for the 8 s EEG in all three phases. Figure 6C presents the Hilbert spectrum, and Figure 6D shows the IMFs of the EEG after applying VMD with the optimized K and PF values. The Fourier spectrogram (multitaper method) for 64 s, including the aforementioned 8 s EEG segment, is shown in Figure 6E. While IMF-1 was almost fixed in the delta wave region, the peak values of the Hilbert spectrum for IMF-2 showed variation across the phases of GA. IMF-2 showed enhanced α waves at approximately 10–12 Hz under propofol-induced GA, as shown in Figure 6E, and shifted forward and backward with the transition to emergence from anesthesia.

Table 2 statistically examines the fluctuations in K, PF, and fitness for each patient during the phase from the anesthetized to awake states, as shown in Figure 5-A, by comparing the mean values before and after awakening. The results indicate significant differences in the K values for Patient #2 and the combined data from three patients but not for the other K or PF values. However, significant differences were detected in the fitness values for all three patients between the means before and after awakening. This suggests that while the K and PF values optimized by GWO are less likely to serve as indicators of the difference between the anesthetized and awake states, the fitness values, which are outputs of the envelope entropy function, may significantly increase during the transition from anesthesia to emergence.

The reasoning here is that if there is no large variation in the envelope entropy as a fitness function, it is acceptable to decompose using a lower K value. A K value of approximately 2 to 3 should suffice for the VMD of EEG signals during GA. Therefore, in the three case studies, K was fixed at 2, PF was fixed at 2000, and PF was optimized to obtain the 10-minute Hilbert spectrum and spectrogram (Figure 6C). It was observed that the α wave enhancement due to propofol-induced GA, as shown in the power spectrogram, corresponded with the separation of the narrow band of α waves represented by IMF-2. We understood that this characteristic α wave enhancement, a feature of propofol-induced GA, could be distinctively extracted in the IMF-2 of VMD.

## 4. Discussion

GA involves three effects: hypnosis, analgesia, and immobility. As anesthesiologists, we want to quantify the hypnotic state of patients; however, there are no other biomarker signals that can quantitatively capture the state of hypnosis apart from EEG, including some evoked potentials. Therefore, there is a focus on quantifying the hypnotic state induced by anesthetic drugs through frequency and amplitude changes in EEGs. Apart from the standard Fourier transform-based power spectral analysis for analyzing the frequency components of EEGs, methods combining mode decomposition with the Hilbert transform, such as the Hilbert–Huang transform method and its variants, have been reported. A disadvantage of Fourier transform-based methods is the inversely proportional relationship between the frequency resolution and temporal resolution, which is affected by the analysis window but is not concerned with the Hilbert transform. As an alternative approach, frequency analysis of EEGs using the continuous wavelet transform has been reported [23]. However, the wavelet transform has drawbacks, such as variations in time and frequency resolution depending on the scale of the basis function and the need to appropriately select the wavelet type and parameters (such as the number of scales and the range of analysis), which requires extensive experience and trial and error.

In this respect, the Hilbert–Huang transform method has the advantage of extracting narrowband signals through mode decomposition. Mode decomposition originates from the Hilbert–Huang transform method combined with the EMD method reported by Huang [11]. There have been several attempts to apply the EMD method to the frequency analysis of EEGs during GA [12,24,25]. Subsequently, methods such as VMD [14] and the empirical wavelet transform (EWT) have been reported [26], along with their combination with the Hilbert transform in variations of the Hilbert–Huang transform. We reported using VMD to analyze EEG signals during GA in 2023 [13]. The EMD method, based on spline interpolation for envelope calculation, is mathematically less robust and is known to be incapable of separating signals into narrow bands. VMD, supported by a more robust mathematical theory based on mathematical optimization, is more effective at separating EEGs into narrower bands and extracting features of their frequency components. However, the necessity for user-defined hyperparameters in VMD requires some form of guidance, which was the theme of this study. It would be interesting to conduct comparative studies to determine which mode decomposition method—EMD, VMD, or even EWT—is superior for analyzing EEGs during GA. However, it is important to emphasize that VMD is dependent on hyperparameters, including the number of decompositions, which is an integral part of the algorithm. Therefore, before comparing different mode decomposition methods, it was necessary for us to introduce the optimization of hyperparameter determination in the current study.

In this analysis, the number of wolves was set to 20 because this number is generally considered appropriate. Increasing the number of wolves might make finding an optimized solution more straightforward, but it can also delay the wolves’ overall convergence (hunting). When we used 20 loops, it was uncertain whether increasing the loop count would lead to overall wolf convergence (hunting). In our situation, convergence of the fitness function is important. Regarding the VMD hyperparameters, the optimization of the K value is crucial, while the PF value, if it has little impact on fitness, may not converge across all wolves and may show variability. However, if the PF value does not affect the fitness value, it might not be necessary to focus on it to a great extent.

VMD itself involves computer analysis based on mathematical optimization theory. Additionally, the K and PF values are selected randomly to calculate the envelope entropy of each IMF obtained through VMD as a fitness value. A single optimization requires the VMD algorithm to be run 800 to 2400 times for K values ranging from 2 to 6, with 20 wolves and 20 loops. This presents a substantial challenge to the speed of the programming language and the computational speed of personal computers, with the analysis speed of the Processing application in Java being insufficient to implement the GWO algorithm in real time. It is, therefore, necessary to consider application development using programming languages with default GPU computation support, such as Julia or Mojo, to achieve more efficient processing.

In this study, we applied the GWO algorithm to optimize the hyperparameters K and PF in VMD, using the envelope entropies of the decomposed IMFs as fitness functions to analyze EEG signals during GA. Our results show that although K and PF varied significantly for each epoch, the envelope entropies remained stable. This implies that in terms of the K value, when the main frequency components for clinical observation are considered, two bands—δ to θ waves and the alpha frequency band, including sleep spindles—are critical during the maintenance phase of GA, suggesting that a K value of 2 or 3 might be sufficient. Subsequently, we considered optimizing the PF value using the GWO algorithm as needed. This study demonstrated the effectiveness of GWO for optimizing hyperparameters in the VMD analysis of biological signals and highlights its promise for future applications.

## 5. Conclusions

In this study, we used the GWO to determine hyperparameters for the VMD method in order to analyze changes in the frequency components of EEG signals during GA. We used the envelope entropy of the IMF decomposed via VMD for the fitness function for optimizing the GWO algorithm. Regarding the decomposition number K for VMD, when a limit of 2 to 6 was set for clinical significance, K varied from 2 to 6 for each epoch of the EEG being analyzed, but, in general, K = 2 or 3, and we believe that the characteristics of the EEG during GA were captured. Regarding the PF value, if the fitness function is optimized to a low value by determining an appropriate K value, the low convergence of the PF value means that different PF values within an allowable range of 10–5000 will not have a significant impact. The determination of appropriate values of VMD hyperparameters by adapting the GWO algorithm provides important findings for the analysis of EEG signals using VMD during GA.

## Figures and Tables

**Figure 1 sensors-24-05749-f001:**
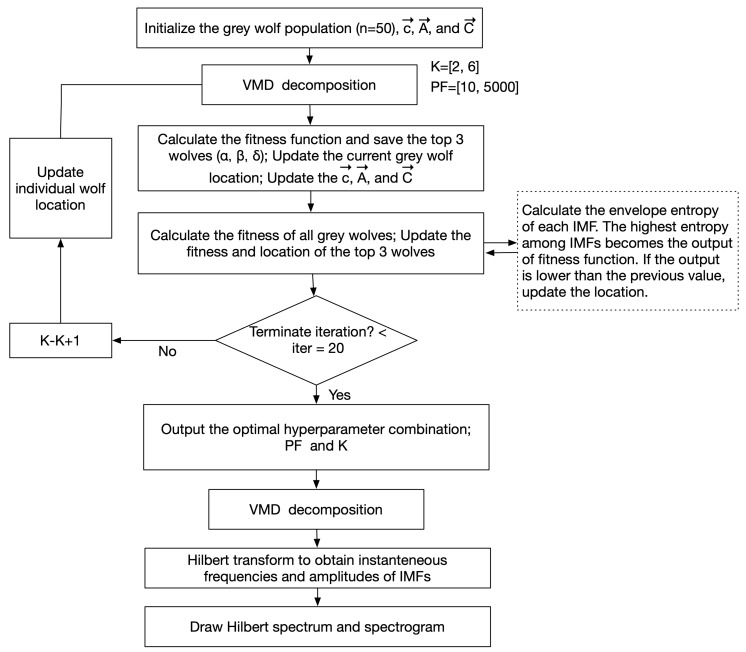
Flowchart of grey wolf optimization (GWO) for variational mode decomposition (VMD).

**Figure 2 sensors-24-05749-f002:**
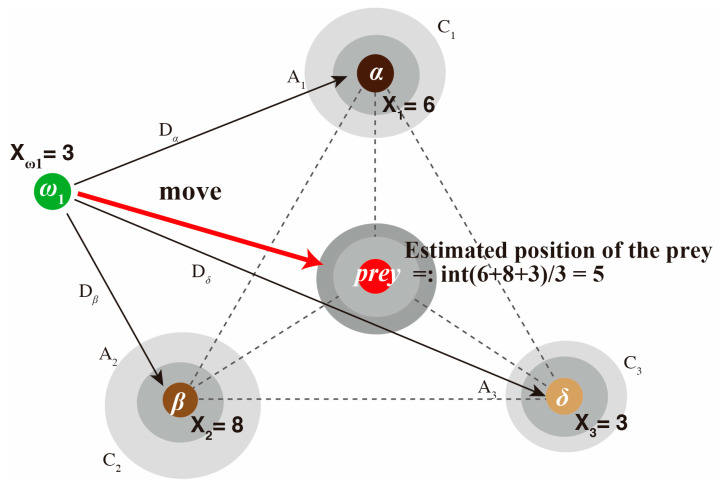
Position updating algorithm for the wolves in the grey wolf optimizer. Three leader/subleader wolves, α, β, and δ, surround the prey, and the position of the prey is inferred from the positions of these three leader wolves, assuming that they surround it. Vectors A and C are coefficient vectors and are calculated for each coordinate. The other wolves in the pack, ω, locate the leader wolves’ positions, which are initially adjusted with a coefficient C and then gradually adjusted in each loop to better approximate the leader wolves’ positions. A random coefficient D is then applied to their distance, allowing them to approach the leader wolves within a range of −1 to +1. As a result, wolves that are closer to the prey than the leader wolves may replace them as the new leaders, enabling the pack to surround the prey more closely. The algorithm specifies three leader wolves and divides the obtained average position by the number of leaders. If the positions of α, β, and δ are 6, 8, and 3, respectively, then the position of the prey would be the midpoint, calculated as (6 + 8 + 3)/3 = 5.7, and the integer 5 becomes the updated position for the prey. The new position of the ω_1_ wolf is adjusted according to the leaders’ positions, taking into account its current position.

**Figure 3 sensors-24-05749-f003:**
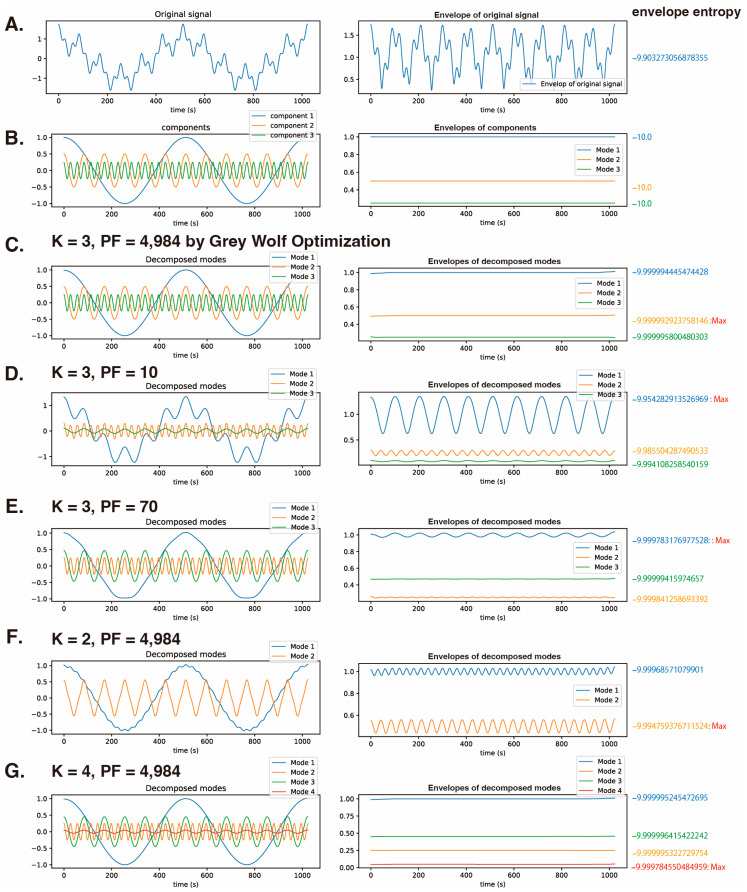
The relationship between the hyperparameters K and PF of VMD and the envelope entropy of each IMF. VMD is applied to a synthetic cosine wave composed of three known frequency components (2, 12, and 36 Hz). $y=cos(2\times 2\pi t)+\frac{1}{2}cos(12\times 2\pi t)+\frac{1}{4}cos(36\times 2\pi t)$ (1024 data points). (**A**) The synthetic cosine wave, its envelope function, and the envelope entropy. (**B**) The three cosine waves, their envelope functions, and envelope entropies. (**C**) Decomposition of the synthetic cosine wave into three IMFs using the GWO algorithm with optimized values of K = 3 and PF = 4984, and calculation of each IMF’s envelope function and envelope entropy. (**D**) Application of VMD with K = 3 and PF = 10, and determination of IMFs, their envelope functions, and envelope entropies. (**E**) Application of VMD with K = 3 and PF = 70, and analysis of IMFs and their envelope functions, and envelope entropies. (**F**) Application of VMD with K = 2 and PF = 4984, and evaluation of the IMFs and their envelope functions, and envelope entropy. (**G**) Application of VMD with K = 4 and PF = 4984, and analysis of the IMFs and their envelope functions, and envelope entropies. K: decomposition number; PF: penalty factor; fitness: fitness value of VMD = the maximum envelope entropy of the IMF.

**Figure 4 sensors-24-05749-f004:**
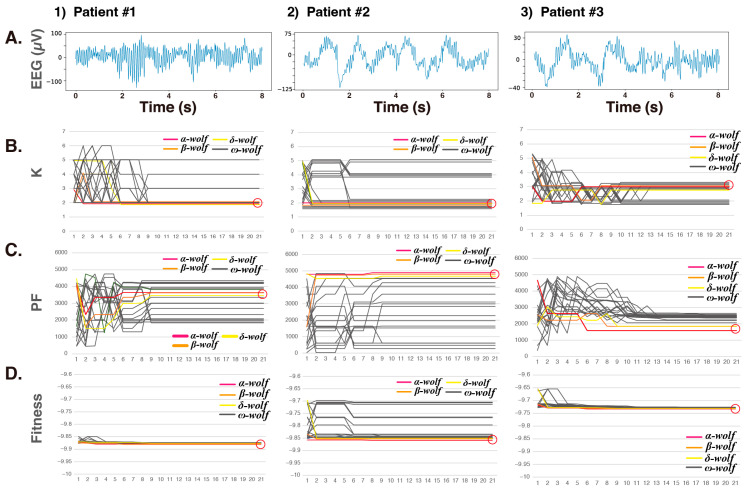
Optimization process for the hyperparameters K and PF of VMD using the GWO algorithm. Twenty wolves and 20 optimization loops, observed over one epoch (8 s, 1024 data points) of EEG during the maintenance phase of GA (obtained through continuous intravenous propofol administration) in three patients. This process involved monitoring the position of each wolf during two-dimensional wolf hunting. (**A**) EEG (8 s). (**B**) The optimization of K. (**C**) The optimization of PF. (**D**) The optimization of fitness. α wolf: red line; β wolf: orange line; δ wolf: yellow line; ω wolves: black lines. K: decomposition number; PF: penalty factor; fitness: fitness value of VMD = the maximum envelope entropy of the IMF.

**Figure 5 sensors-24-05749-f005:**
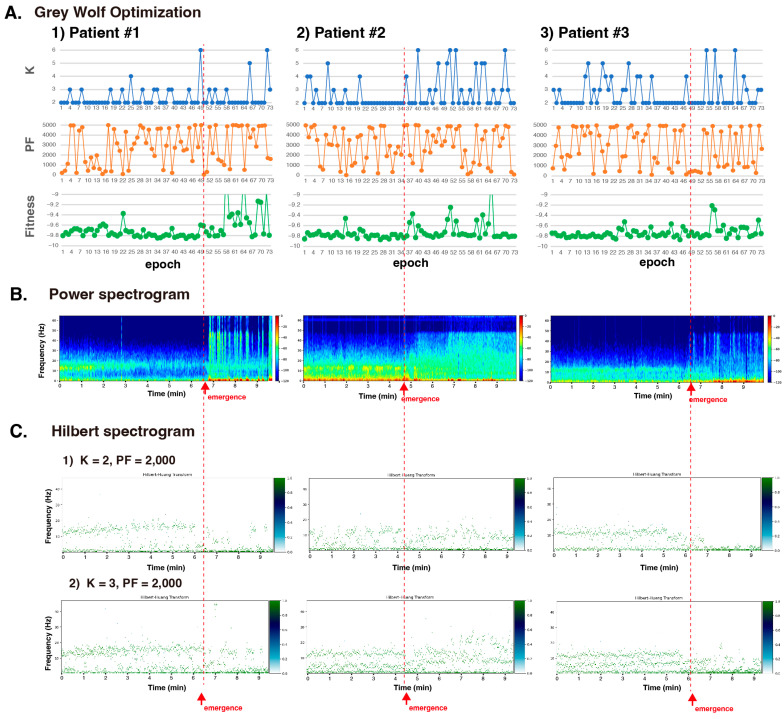
Grey wolf optimization in the VMD of EEGs obtained from three patients of GA induced by continuous intravenous propofol. (**A**) The optimal solutions for K and PF were determined through fitness function optimization using the GWO algorithm. Twenty wolves and 20 optimization loops for all 73 epochs. Each epoch lasted for 8 s (128 Hz, 1024 data points) of the 10-minute period before and after awakening from GA. Additionally, K was set in the range of 2 to 6 (in increments of 1), and the PF values ranged from 10 to 5000 (in increments of 10). (**B**) The spectrogram for the 10-minute interval was obtained using the multitaper method and is displayed in Figure 4B. (**C**) Using VMD with K = 2 or 3 and PF = 2000, the signal was decomposed into IMFs and a Hilbert spectrogram was obtained. The gradient color bar shows the relative power value of the signal. K: decomposition number; PF: penalty factor; fitness: fitness value of VMD = the maximum envelope entropy of the IMF.

**Figure 6 sensors-24-05749-f006:**
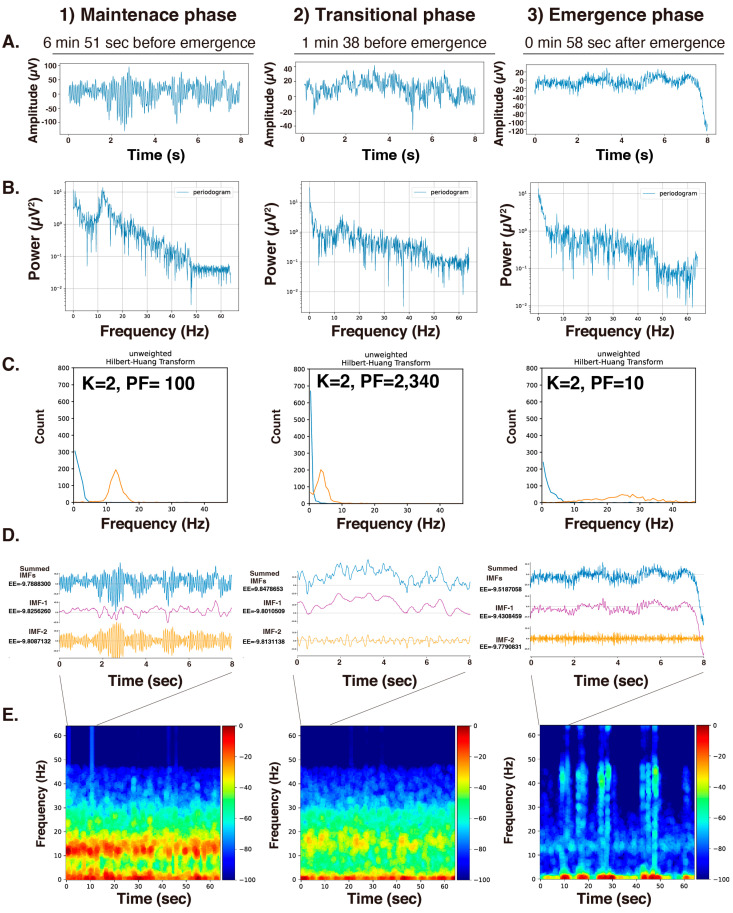
Analysis of 8 s EEG segments during the (1) maintenance, (2) transition, and (3) emergence phases of general anesthesia induced by propofol. (**A**) Original EEG (8 s, 128 Hz, 1024 data points). (**B**) The power spectrum obtained through Fourier analysis. (**C**) Hilbert spectrum. To optimize K and PF, GWO analysis was conducted with 20 wolves, 20 repetitions, and settings of K = 2 to 6 (in increments of 1) and PF = 10 to 5000 (in increments of 10). (**D**) Decomposed intrinsic mode functions (IMFs). (**E**) The power spectrogram for a 64 s period was determined using the multitaper method. The first 8 s were the subject of the VMD analysis. The gradient color bar shows the relative power value of the signal. K: decomposition number; PF: penalty factor; fitness: fitness value of VMD = the maximum envelope entropy of the IMF.

**Table 1 sensors-24-05749-t001:** Convergence rate in the optimization process for the hyperparameters K and PF of VMD using the GWO algorithm.

		K				PF				Fitness			
		Patient				Patient				Patient			
		#1	#2	#3	Mean ± SD	#1	#2	#3	Mean ± SD	#1	#2	#3	Mean ± SD
Converge value		2.35	2.7	2.8	2.6 ± 0.2	3178	2918	2315	2804 ± 443	−9.8769	−9.8156	−9.7289	−9.8071 ± 0.07
Iteration		%				%				%			
	1	143	117	133	131 ± 13	78	73	108	86 ± 19	99.9248	99.9940	99.7606	99.8361 ± 0.0829
	2	149	113	120	127 ± 19	82	80	120	94 ± 23	99.8326	99.8327	99.7607	99.8562 ± 0.0662
	3	143	111	104	119 ± 21	96	82	145	108 ± 33	99.8325	99.8346	99.8927	99.8946 ± 0.0610
	4	149	111	**100**	120 ± 26	96	82	138	106 ± 29	99.8346	99.8346	99.9371	99.9141 ± 0.0610
	5	143	111	96	117 ± 24	100	90	144	111 ± 29	99.9729	99.8346	99.9370	99.9200 ± 0.0708
	6	121	**100**	96	106 ± 14	91	89	133	104 ± 24	99.9902	99.9881	99.9370	99.9802 ± 0.0678
	7	121	**100**	93	105 ± 15	91	89	130	104 ± 23	99.9902	99.9881	99.9677	99.9820 ± 0.0155
	8	106	**100**	95	100 ± 6	97	95	125	106 ± 17	99.9939	99.9909	99.9733	99.9860 ± 0.0125
	9	**100**	**100**	93	98 ± 4	**100**	**100**	118	106 ± 10	**100.000**	**100.000**	99.9733	99.9945 ± 0.0111
	10	**100**	**100**	93	98 ± 4	**100**	**100**	112	104 ± 7	**100.000**	**100.000**	99.9910	99.9970 ± 0.0096
	11	**100**	**100**	98	99 ± 1	**100**	**100**	103	101 ± 2	**100.000**	**100.000**	99.9970	99.9990 ± 0.0052
	12	**100**	**100**	98	99 ± 1	**100**	**100**	103	101 ± 2	**100.000**	**100.000**	99.9976	99.9992 ± 0.0017
	13	**100**	**100**	**100**	**100 ± 0**	**100**	**100**	101	100 ± 1	**100.000**	**100.000**	99.9986	99.9995 ± 0.0014
	14	**100**	**100**	**100**	**100 ± 0**	**100**	**100**	101	100 ± 1	**100.000**	**100.000**	99.9988	99.9996 ± 0.0008
	15	**100**	**100**	**100**	**100 ± 0**	**100**	**100**	101	100 ± 1	**100.000**	**100.000**	99.9988	99.9996 ± 0.0007
	16	**100**	**100**	**100**	**100 ± 0**	**100**	**100**	101	**100 ± 0**	**100.000**	**100.000**	99.9989	99.9996 ± 0.0007
	17	**100**	**100**	**100**	**100 ± 0**	**100**	**100**	101	**100 ± 0**	**100.000**	**100.000**	99.9990	99.9997 ± 0.0006
	18	**100**	**100**	**100**	**100 ± 0**	**100**	**100**	**100**	**100 ± 0**	**100.000**	**100.000**	**100.000**	**100.0000 ± 0.000**
	19	**100**	**100**	**100**	**100 ± 0**	**100**	**100**	**100**	**100 ± 0**	**100.000**	**100.000**	**100.000**	**100.0000 ± 0.000**
	20	**100**	**100**	**100**	**100 ± 0**	**100**	**100**	**100**	**100 ± 0**	**100.000**	**100.000**	**100.000**	**100.0000 ± 0.000**

Bold numbers indicate reaching the convergence value (=100%). K: decomposition number; PF: penalty factor; fitness: fitness value of VMD = the maximum envelope entropy of the IMF; SD: standard deviation.

**Table 2 sensors-24-05749-t002:** Grey wolf optimization in the VMD of EEGs obtained from three patients of general anesthesia induced by continuous intravenous propofol.

	K			PF			Fitness		
	Before Emergence	After Emergence		Before Emergence	After Emergence		Before Emergence	After Emergence	
Patient	Mean ± SD	Mean ± SD	*p*-Value	Mean ± SD	Mean ± SD	*p*-Value	Mean ± SD	Mean ± SD	*p*-Value
#1	2.3 ± 0.7	2.5 ± 1.0	0.525	2722 ± 1792	3190 ± 1914	0.309	−5.7779 ± 0.0898	−9.4826 ± 0.4004	<0.001 *
#2	2.4 ± 0.8	3.0 ± 1.4	0.014 *	2881 ± 1678	3071 ± 1772	0.675	−9.7762 ± 0.0728	−9.6799± 0.2341	0.016 *
#3	2.6 ± 1.0	2.8 ± 1.4	0.624	2271 ± 1834	2271 ± 1922	0.107	−9.7607 ± 0.0714	−9.6886 ± 0.1563	0.008 *
Mean	2.4 ± 0.8	2.7 ± 1.3	0.016 *	2625 ± 1768	2844 ± 1869	0.989	−8.4383 ± 0.0780	−9.6170± 0.2636	<0.001 *
SD	0.2 ± 0.1	0.3 ± 0.2		316 ± 81	500 ± 84		2.3039 ± 0.0103	0.1165 ± 0.1247	

The optimal solutions for the K, PF, and fitness values were determined through fitness function optimization using the GWO algorithm (20 wolves and 20 optimization loops for all 73 epochs). The values between before-emergence and after-emergence were statistically compared. K: decomposition number; PF: penalty factor; fitness: fitness value of VMD = the maximum envelope entropy of the IMF; SD: standard deviation. * *p* < 0.05 between before emergence and after emergence.

## Data Availability

The EEG dataset used is available on the author’s GitHub site (https://github.com/teijisw/EEG_DataSet 4 July 2024) [18]. The programming codes used in the analysis in this paper are available as Appendix A.

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
