# Peer review of "Variational Mode Decomposition Analysis of Electroencephalograms during General Anesthesia: Using the Grey Wolf Optimizer to Determine Hyperparameters"

_sensors, 2024, doi:10.3390/s24175749_

Round 1

Reviewer 1 Report

Comments and Suggestions for Authors

1. The contributions and the hypothesis should be stated more clearly. The structure of the proposed algorithm should be explained.

2. The reason for choosing the grey wolf optimizer should be stated. There are many kinds of swarm intelligence algorithms. How about other algorithms?

3. More explanation should be given for Fig.1. The meaning of 4->8,4->3 should be stated.

Comments on the Quality of English Language

None

Author Response

Comments 1: The contributions and the hypothesis should be stated more clearly. The structure of the proposed algorithm should be explained.

Response 1: 1) As the reviewer suggests, we added the hypothesis in the section of introduction, as follow:

Page 2, line 91-page 3, line 103: As the effects of GA drugs diminish, there are changes in the frequency components of EEG signals. In such dynamically changing EEG signals, it is assumed that the hyperparameters of VMD also need to change dynamically. Based on this hypothesis, we conducted an experiment where the hyperparameters of VMD were calculated in real time for each EEG analysis epoch to optimize them. Specifically, during the approximately 10 minutes from the maintenance of GA to awakening, we observed how VMD's hyperparameters were determined via the GWO algorithm and how the associated VMD process was optimized. We also examined whether the extraction of EEG features using VMD could lead to the estimation of anesthesia depth through the acquisition of IMF components and the analysis of their characteristic frequency components. In this study, we report on the use of an optimized VMD method for decomposing EEG data recorded under GA, demonstrating how it can be used to acquire a series of IMF components and examining its utility.

2) We newly added Figure 1 (Page 5) for the flowchart of grey wolf optimization (GWO) for variational mode decomposition (VMD) to explain the structure of proposed algorithm

Comments 2: The reason for choosing the grey wolf optimizer should be stated. There are many kinds of swarm intelligence algorithms. How about other algorithms?

Response 2: We have not tested other Swarm algorithms, but we have included the following description in the Introduction section on why we chose GWO for this study:

Page 2, line 80-line 91: The signals analyzed are complex and volatile, making it challenging to determine the two parameters influencing them. Therefore, selecting appropriate parameters is key to analyzing signal data through VMD. Attempts to optimize VMD using intelligent algorithms have been reported in various fields. It is essential to use an algorithm that is easy to implement, has an adaptive convergence coefficient and information feedback mechanisms, and balances local optimization with global search. The GWO algorithm, which mimics the hunting behavior and leadership hierarchy of grey wolves, has been successful in many real-world applications. In the same number of iterations, the GWO algorithm has been reported to converge significantly better than other algorithms. Therefore, in this study, we implemented the GWO algorithm in the EEG analysis process with the aim of optimizing VMD. To do this, we used envelope entropy as the fitness function of the GWO algorithm to optimize the number of K modes and the PF in EEG signals during GA.

Comments 3: More explanation should be given for Fig.1. The meaning of 4->8,4->3 should be stated.

Response 3: We have simplified the content of Figure 2 and added a more detailed explanation to the legend of Figure 2 as follows:

Page 6, Figure 2, legend: Position updating algorithm for the wolves in the grey wolf optimizer. Three leader/subleader wolves, α, β, and δ, surround the prey, and the position of the prey is inferred from the positions of these three leader wolves, assuming that they surround it. Vectors A and C are coefficient vectors and are calculated for each coordinate. The other wolves in the pack, ω, locate the leader wolves’ positions, which are initially adjusted with a coefficient C and then gradually adjusted in each loop to better approximate the leader wolves’ positions. A random coefficient D is then applied to their distance, allowing them to approach the leader wolves within a range of −1 to +1. As a result, wolves that are closer to the prey than the leader wolves may replace them as the new leaders, enabling the pack to surround the prey more closely. The algorithm specifies three leader wolves and divides the obtained average position by the number of leaders. If the positions of α, β, and δ are 6, 8, and 3, respectively, then the position of the prey would be the midpoint, calculated as (6+8+3)/3 = 5.7, and the integer 5 becomes the updated position for the prey. The new position of the ω1 wolf is adjusted according to the leaders' positions, taking into account its current position.

Reviewer 2 Report

Comments and Suggestions for Authors

The peer-reviewed article is devoted to the actual and important scientific problem. The article can be published in Sensors after some editing in accordance with the following remarks.

- The introduction to the article does not contain a review of the issues of application of various methods of diagnostics of patients' states based on EEG signals during anesthesia. This should be corrected by expanding this section.

- In the Discussion, the authors discuss comparison with Fourier analysis, but do not discuss comparison with wavelet analysis, which would be reasonable.

- The authors should provide a general table with numerical data illustrating the effectiveness of their algorithm.

Author Response

Comments 1: The introduction to the article does not contain a review of the issues of application of various methods of diagnostics of patients' states based on EEG signals during anesthesia. This should be corrected by expanding this section.

Response 1: In accordance with the reviewer's suggestion, we have introduced references concerning reviews of EEG measurements during anesthesia, new techniques, and the current issues with commercial monitoring technologies, and have added these descriptions to the Introduction section, as follows:

Page 1, line 38-page 2, line 51: There have been numerous reports on new technologies for EEG signal processing that can significantly broaden the scope of EEG monitoring for assessing patients' anesthetic sedation states during GA [4, 5]. In addition, there are growing expectations for advancements in the intelligent sensors that are involved in detecting biological signals, spurred by the rapid growth of flexible electronics [6, 7]. The potential benefits of these technologies, when combined with fast AI diagnostic algorithms, are significant. This combination holds promise for more reliable clinical diagnostics, especially in the analysis of EEGs. However, the analysis parameters of commercially available EEG monitors have only been optimized for certain inhaled anesthetics and propofol, and other anesthetics or the effects of age have not been accounted for [8-10]. Additionally, each algorithm is proprietary and not disclosed, which limits comparative studies and further developments. Therefore, establishing new technologies for effective feature extraction of EEGs during GA using open algorithms is crucial for anesthesiologists who aim to provide safe and effective anesthesia.

  • Chaddad A, Wu Y, Kateb R, Bourid A. Electroencephalography signal processing: a comprehensive review and analysis of methods and techniques. Sensors2023, 23(14), 6434. doi:10.3390/s23146434
  • Sharma R, Meena Emerging trends in EEG signal processing: a systematic review. SN Comp Sci 2024, 5: 415. doi:10.1007/s42979-024-02773-w
  • Sun Y, Wei C, Cui V, Xiu M, Wu A. Electroencephalography: clinical applications during the perioperative period. Front Med 2020, 7:251. doi: 10.3389/fmed.2020.00251
  • Yuan I, Xu T, Kurth CD. Using electroencephalography (EEG) to guide propofol and sevoflurane dosing in pediatric anesthesia. Anesthesiol Clin 2020 38(3):709-725. doi: 10.1016/j.anclin.2020.06.007
  • Schultz B, Schultz M, Boehne M, Dennhardt N. EEG monitoring during anesthesia in children aged 0 to 18 months: amplitude-integrated EEG and age effects. BMC Pediatr 2022, 26;22(1):156. doi: 10.1186/s12887-022-03180-x.

Comments 2: In the Discussion, the authors discuss comparison with Fourier analysis, but do not discuss comparison with wavelet analysis, which would be reasonable.

Response 2: In line with the reviewer's suggestion, we have added the following passage about the continuous wavelet method at the beginning of the discussion section, and cited the relevant paper as a reference.

Page 14, line 389-line 394: As an alternative approach, frequency analysis of EEGs using the continuous wavelet transform has been reported [23]. However, the wavelet transform has drawbacks, such as variations in time and frequency resolution depending on the scale of the basis function and the need to appropriately select the wavelet type and parameters (such as the number of scales and the range of analysis), which requires extensive experience and trial and error.

  • Mousavi SM, AdamoÄŸlu A, Demiralp T,  Shayesteh MG. A wavelet transform based method to determine depth of anesthesia to prevent awareness during general anesthesia. Comput Math Methods Med 2014, 2014:354739. doi: 10.1155/2014/354739. 

Comments 3: The authors should provide a general table with numerical data illustrating the effectiveness of their algorithm.

Response 3: In accordance with the reviewer's suggestion, we have added new Tables 1 and 2 that present numerical data. Table 1 (in page 11) shows the convergence rate in the optimization process for the hyperparameters K and PF of VMD using the GWO algorithm, while Table 2 (in page 14) statistically compares the optimal solutions for K, PF, and fitness values before and after awakening from general anesthesia induced by continuous intravenous propofol, titled "Grey wolf optimization in VMD of EEG obtained from three patients.

Reviewer 3 Report

Comments and Suggestions for Authors

This manuscript needs reconsideration after a major revision. The authors should resolve following questions to further improve the manuscript.

1. The authors stated that it is great challenge for obtaining clear waveforms. Recently, the fast development of flexible electronics play crucial roles in intelligent healthcare. For instance, there is a flexible and reliable pulse sensor for pulse waves detection and intelligent diagnostics of CVD events [Cell Reports Physical Science 4, 101690]. Could the authors discuss these in the introduction section? And discussed whether there was a possibility of flexible electronics for EEG reliable monitoring.

2. in line 15, the full name of VMD should be provided.

3. From the EEG waveforms, what’s the physiological indicators reflecting general anesthesia? Heart rate? Or any other indicators? Furthermore, how about the indicators’ variation in these three patients?

4. The authors should compare their results with existing studies. This may highlight the advantages of this study.

Author Response

Comments 1: The authors stated that it is great challenge for obtaining clear waveforms. Recently, the fast development of flexible electronics play crucial roles in intelligent healthcare. For instance, there is a flexible and reliable pulse sensor for pulse waves detection and intelligent diagnostics of CVD events [Cell Reports Physical Science 4, 101690]. Could the authors discuss these in the introduction section? And discussed whether there was a possibility of flexible electronics for EEG reliable monitoring. 

Response 1: Thank you for enlightening us about the new sensor technologies. Along with two new references, we have added the following descriptions of these technologies to the Introduction.

Page 1, line 38-page 2, line 44: There have been numerous reports on new technologies for EEG signal processing that can significantly broaden the scope of EEG monitoring for assessing patients' anesthetic sedation states during GA [4, 5]. In addition, there are growing expectations for advancements in the intelligent sensors that are involved in detecting biological signals, spurred by the rapid growth of flexible electronics [6, 7]. The potential benefits of these technologies, when combined with fast AI diagnostic algorithms, are significant. This combination holds promise for more reliable clinical diagnostics, especially in the analysis of EEGs.

  • Ma Z , Hua H,  You C, Ma Z, Guo W, Yang X, Qiu S, Zhao N, Y Z, Ho D, Yan BP, Khoo BL. FlexiPulse: A machine-learning-enabled flexible pulse sensor for cardiovascular disease diagnostics. Cell Rep Phys Sci 2023, 4:101690. doi:10.1016/j.xcrp.2023.101690
  • Velcescu A, Lindley A, Cursio C, Krachunov S, Beach C, Brown CA, Jones AKP, Casson AJ. Flexible 3D-printed EEG electrodes. Sensors201919(7), 1650. doi:10.3390/s19071650

Comments 2:. in line 15, the full name of VMD should be provided. 

Response 2: We fixed it.

Comments 3: From the EEG waveforms, what’s the physiological indicators reflecting general anesthesia? Heart rate? Or any other indicators? Furthermore, how about the indicators’ variation in these three patients? 

Response 3: General anesthesia involves three effects: hypnosis, analgesia, and immobility. As anesthesiologists, we want to quantify the hypnotic state of patients, but there are no other biomarker signals that can quantitatively capture the state of hypnosis apart from electroencephalography (EEG), including some evoked potentials. For instance, heart rate and blood pressure during general anesthesia are used to assess the state of analgesia but are problematic for evaluating hypnosis. Therefore, there is a focus on quantifying the hypnotic state induced by anesthetic drugs through changes in EEG. We have discussed this point at the beginning of the Discussion section as follows:

Page 13, line 379- page 14, line 383: GA involves three effects: hypnosis, analgesia, and immobility. As anesthesiologists, we want to quantify the hypnotic state of patients; however, there are no other biomarker signals that can quantitatively capture the state of hypnosis apart from EEG, including some evoked potentials. Therefore, there is a focus on quantifying the hypnotic state induced by anesthetic drugs through frequency and amplitude changes in EEGs.

Comments 4: The authors should compare their results with existing studies. This may highlight the advantages of this study.

Response 4: As you pointed out, it is indeed important to compare different methods of mode decomposition. We have previously published papers on the application of Empirical Mode Decomposition (EMD) and Variational Mode Decomposition (VMD) to EEG during general anesthesia (references 12 and 13). We are currently advancing our research on mode decomposition methods using Wavelet transform. However, we have realized that it is crucial to prioritize optimizing the determination of hyperparameters for VMD in the current study, which has necessitated prioritizing the preparation of this paper. We have added the following to the Discussion to address this matter:

Page 15, line 408-line 414: It would be interesting to conduct comparative studies to determine which mode decomposition method—EMD, VMD, or even EWT—is superior for analyzing EEGs during GA. However, it is important to emphasize that VMD is dependent on hyperparameters, including the number of decompositions, which is an integral part of the algorithm. Therefore, before comparing different mode decomposition methods, it was necessary for us to introduce the optimization of hyperparameter determination in the current study.

Round 2

Reviewer 3 Report

Comments and Suggestions for Authors

This manuscript is suitable for publication.